# Machine learning applied to enzyme turnover numbers reveals protein structural correlates and improves metabolic models

David Heckmann[1], Colton J. Lloyd[1], Nathan Mih[1], Yuanchi Ha[1], Daniel C. Zielinski[1], Zachary B. Haiman[1], Abdelmoneim Amer Desouki[2], Martin J. Lercher [2] & Bernhard O. Palsson [1,3]

Knowing the catalytic turnover numbers of enzymes is essential for understanding the growth rate, proteome composition, and physiology of organisms, but experimental data on enzyme turnover numbers is sparse and noisy. Here, we demonstrate that machine learning can successfully predict catalytic turnover numbers in *Escherichia coli* based on integrated data on enzyme biochemistry, protein structure, and network context. We identify a diverse set of features that are consistently predictive for both in vivo and in vitro enzyme turnover rates, revealing novel protein structural correlates of catalytic turnover. We use our predictions to parameterize two mechanistic genome-scale modelling frameworks for proteome-limited metabolism, leading to significantly higher accuracy in the prediction of quantitative proteome data than previous approaches. The presented machine learning models thus provide a valuable tool for understanding metabolism and the proteome at the genome scale, and elucidate structural, biochemical, and network properties that underlie enzyme kinetics.

[1] Department of Bioengineering, University of California, San Diego, La Jolla, CA 92093–0412, USA. [2] Institute for Computer Science and Department of Biology, Heinrich Heine University, 40225 Düsseldorf, Germany. [3] The Novo Nordisk Foundation Center for Biosustainability, Technical University of Denmark, 2800 Lyngby, Denmark. Correspondence and requests for materials should be addressed to D.H. (email: dheckmann@ucsd.edu) or to B.O.P. (email: palsson@ucsd.edu)

In order to prevail in a given environment, living cells have to allocate a finite amount of protein into diverse cellular functions. Understanding optimal global proteome allocation is a central problem in systems biology and underlies important cellular properties like growth rate[1], thermosensitivity[2], and overflow metabolism[3]. A central goal of computational biology is to develop the ability to predict the genome-scale proteome allocation that leads to the highest fitness—or, as a proxy, growth rate—subject to a given environment and protein budget.

Traditional approaches like flux balance analysis (FBA)[4] search for the optimal growth rate that can be achieved given a set of uptake fluxes and metabolic network stoichiometric constraints, but do not account for the protein allocation problem. In order to extend FBA accordingly, a variety of genome-scale models (GEMs) of metabolism have been developed that consider the cost of expressing metabolic enzymes. Some GEMs extend the approach of FBA with an additional constraint on the total amount of protein the cell has available to catalyze metabolic fluxes that maximize cell growth[5–7]. Other more detailed GEMs include the entire gene expression machinery to explicitly model the proteome composition as a consequence of translation rates and growth-dependent dilution of macromolecules to daughter cells[8–11].

In all of these modelling approaches, the protein cost that arises from achieving a certain flux through a reaction is determined by the catalyzing enzyme's effective turnover rate, $k_{eff}$ (also called apparent turnover rate, $k_{app}$). Thus, GEMs that account for proteome allocation rely heavily on estimates of effective turnover rates. In the past, these estimates were either obtained by random sampling[7], parameter fitting[12,13], or, in most cases, by using in vitro measurements of enzyme turnover numbers, $k_{cat}$[5,6]. In theory, in vitro $k_{cat}$ measurements should provide a reasonable upper limit on $k_{eff}$, where incomplete saturation, thermodynamic effects, posttranslational modifications, and allosteric regulation will render $k_{eff}$ in vivo lower than $k_{cat}$ in vitro[14]. Nevertheless, in practice, in vitro assays of enzyme activity are sensitive to a variety of extraction and assay parameters, leading to noisy estimates and rendering large-scale estimation of $k_{cat}$ in vitro difficult (see Bar-Even et al.[15] for discussion). To address this issue and to provide estimates of $k_{eff}$ in vivo, proteomic data across diverse growth conditions was recently combined with in silico flux predictions to calculate $k_{app,max}$, the maximal $k_{eff}$ across conditions[14]. This in vivo estimate is a promising candidate for

parameterization of all GEMs that account for enzyme kinetics. Nevertheless, the scope of datasets on both in vitro $k_{cat}$ and $k_{app,max}$ is far from genome-scale, with a coverage of direction-specific reactions in E. coli of about 12% for $k_{cat}$ in vitro and 8% for $k_{app,max}$ (Supplementary Figure 2).

It would thus be desirable to understand the underlying genome-scale patterns of catalytic enzyme turnover rates—a major part of the kinetome[16]—and thus protein efficiency. For in vitro $k_{cat}$, global trends were found in relation to the basic biochemical mechanism of the reaction, measured as the first digit of the respective EC numbers[15]. In addition to EC numbers, enzyme molecular weight and reaction flux were shown to correlate with $k_{cat}$ in vitro[5,15], indicating that differential selection pressure explains variance in turnover numbers[17]. It is unclear how these features act together to explain variance in $k_{cat}$. Machine learning (ML) methods for the development of complex statistical models have been successfully applied to modelling bacterial physiology[18–20], enzyme specificity[21,22], and enzyme affinity[21,23], with applications in metabolic engineering and synthetic biology[24,25]. Here, we combine known correlates of $k_{cat}$ with novel features for enzyme structure, biochemical mechanism, network context, and assay conditions to build ML models of $k_{cat}$ in vitro and $k_{app,max}$ that can predict these parameters at the genome scale. Application of these ML models to the parameterization of mechanistic GEMs enables improved predictions of proteome allocation.

## Results

**Compiling features for machine learning.** In order to build predictive ML models of enzyme catalytic turnover rates, we compiled a diverse set of features that include network properties, enzyme structural properties, biochemical mechanism information, and assay conditions (Fig. 1, details in Methods and Supplementary Table 2).

Network properties were extracted from a GEM of E. coli K-12 MG1655, iML1515[26]: The average flux across diverse growth conditions was obtained with a Monte Carlo sampling approach and parsimonious FBA[27] (see Methods). The propensity of an enzyme component to participate in multiple reactions—the generalist property—was in the past found to be associated with lower catalytic turnover rates[28]. We thus quantified the tendency of an enzyme to catalyze multiple reactions from the gene-protein-reaction (GPR) rules of iML1515. Furthermore, the

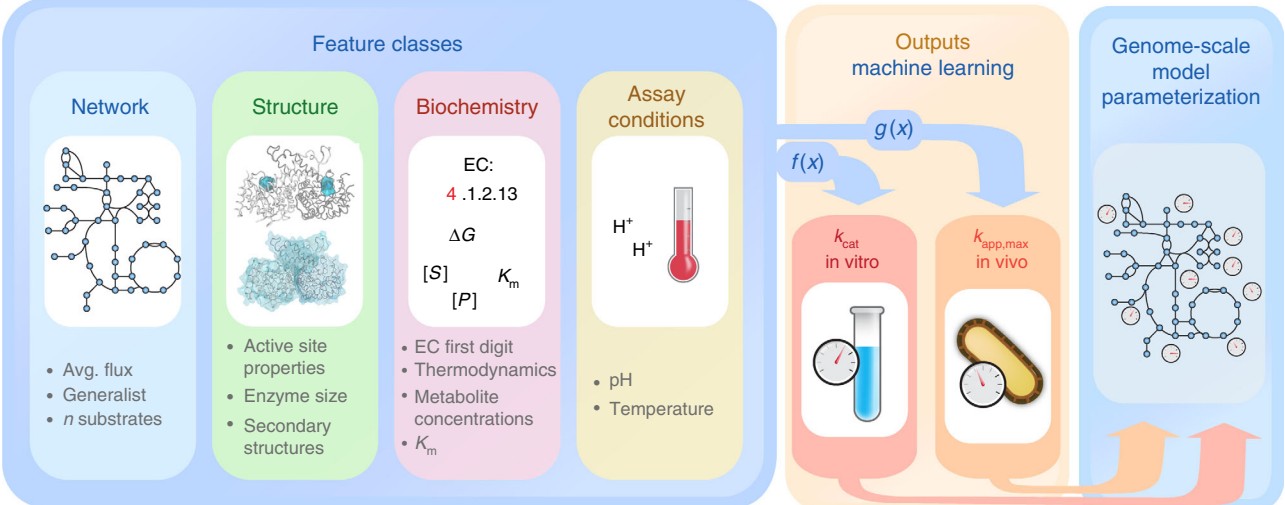

**Fig. 1** Machine learning of catalytic turnover numbers for genome-scale metabolic model (GEM) parameterization. A feature set from diverse classes is curated and mapped to independently build machine learning (ML) models of both $k_{cat}$ in vitro ($f(x)$) and $k_{app,max}$ in vivo ($g(x)$). The inferred ML models are used to predict $k_{cat}$ in vitro or $k_{app,max}$ at the genome-scale to parameterize GEMs

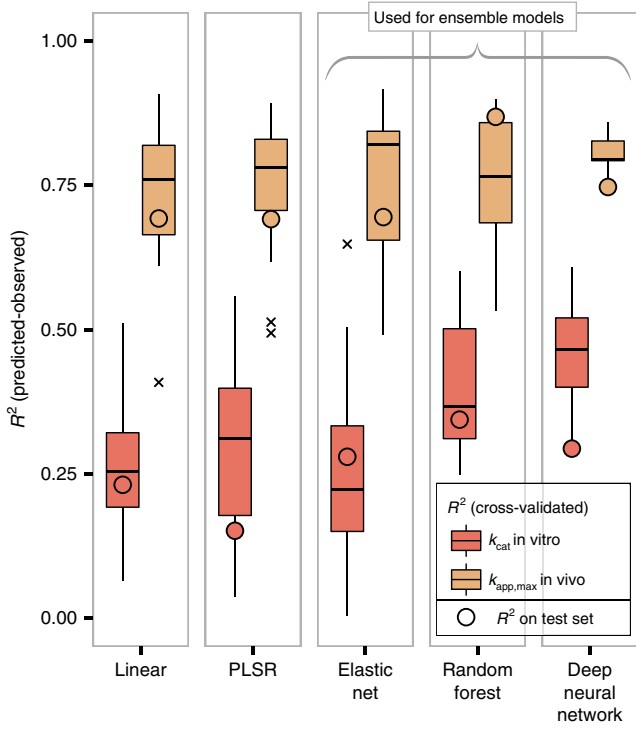

**Fig. 2** Machine learning model performances for $k_{app,max}$ and $k_{cat}$ in vitro. Center lines show the median $R^2$ across five times repeated five-fold cross-validation (25 validations), except for the deep learning case, where the median for a single round of five-fold cross-validation (five validations) is shown. Box limits represent the 1st and 3rd quartiles, whiskers extend to values that lie within the 1.5x interquartile range, and the remaining points are shown as outliers (marked x). Circles show $R^2$ for a test set consisting of 20% of the available samples that were not used for hyperparameter optimization. This resulted in a training set of 172 observations of $k_{cat}$ in vitro and 106 observations of $k_{app,max}$. For the test set, 43 and 27 observations were used for $k_{cat}$ in vitro and $k_{app,max}$, respectively. See Methods for details on hyperparameter optimization

number of enzyme substrates was extracted from the stoichiometric matrix of *i*ML1515.

We hypothesized that the structural properties of enzymes contain information on catalytic turnover constants. To this end, we extracted enzyme structural properties from protein structures in the Protein Data Bank[29] and homology models from the I-TASSER modelling pipeline[30,31] (see Methods). Global structural disorder and molecular weight were used as ML model features. The relative occurrence of secondary structures classes are highly correlated with the fraction of structural disorder, and we decided not to include them in the ML model to avoid co-linear features. We further expected properties of the catalytic site structure to be particularly informative about enzyme turnover and thus extracted catalytic site information from the Catalytic Site Atlas[32]. In particular, we used active site depth, active site solvent exposure, active site hydrophobicity, the number of residues contributing to the active site, and active site secondary structure as model features (see Supplementary Table 1 for details).

Further information on enzyme biochemistry was included in the form of EC numbers, thermodynamic efficiency, Michaelis constants ($K_m$s), and metabolite concentrations (see Methods).

For ML models of in vitro $k_{cat}$s we included assay pH and assay temperature as model features to correct for these assay conditions.

As no convincing correlation between the properties of enzyme substrate structural properties and in vitro $k_{cat}$ was found previously[15], we decided not to include substrate structural properties as features.

**Compiling output data for machine learning**. Traditionally, enzyme catalytic turnover numbers are measured in biochemical in vitro assays, a quantity we refer to as $k_{cat}$ in vitro. We extracted information on $k_{cat}$ in vitro for *E. coli* from the BRENDA[33], SABIO-RK[34], and Metacyc[35] databases (Supplementary Figure 1). These extracted values were filtered to avoid non-wild type enzymes, non-physiological substrates, and redundancy across databases (see Methods for details). In addition to in vitro measurements, we used in vivo estimates of effective enzyme turnover, $k_{app,max}$, that were obtained as the maximum effective turnover rate across diverse growth conditions[14]. The final data set has 215 complete observations—i.e., all features and output are available—for $k_{cat}$ in vitro and 133 complete observations for $k_{app,max}$ (Supplementary Figure 2); as discussed below, this set can be extended through imputation of selected features, yielding 497 and 234 complete observations for $k_{cat}$ in vitro and $k_{app,max}$, respectively.

**Training predictive models of enzyme turnover numbers**. We utilized the compiled feature set to separately train ML models for $k_{cat}$ in vitro and $k_{app,max}$ (Fig. 1). A diverse set of regression algorithms was trained using repeated five-fold cross-validation (see Methods and Supplementary Table 2). We find that the choice of algorithm has only a small effect on model performance, where the mean cross-validated $R^2$ between predictions and validation tends to be smaller in linear modelling techniques (linear regression, PLSR, elastic net) as compared to the more complex models (random forest and deep neural network) for the $k_{cat}$ in vitro models (Fig. 2). The predictive performance of the models is significantly higher for $k_{app,max}$ than for $k_{cat}$ in vitro, showing average cross-validated $R^2$s of 0.76 and 0.31, respectively (Fig. 2, see Supplementary Figure 3 for root mean squared errors (RMSEs)). Model performance estimation through cross validation can be positively biased because hyperparameters are optimized in the process, but using an independent test set confirms our findings (Fig. 2). We thus expect models of $k_{app,max}$ to be more suitable for predicting catalytic turnover rates at the genome scale.

**Models exhibit similarity in feature importance**. Although ML models of $k_{app,max}$ achieved a higher prediction accuracy than those for $k_{cat}$ in vitro, both models are able to explain significant variance in catalytic turnover rates from our feature set. Which features contribute most to these predictions? We analyzed feature importance in the random forest models by examining the average increase in mean squared error that results from randomly permuting a respective feature vector across 500 trained decision trees (Fig. 3).

We find that feature importance is significantly correlated between models for $k_{cat}$ in vitro and in vivo $k_{app,max}$ (Spearman Rank correlation 0.46, $p < 0.025$, $n = 24$, $S = 1214$, see Methods). In silico flux is the most important feature for both in vitro $k_{cat}$ and in vivo $k_{app,max}$, confirming the hypothesized significant role of evolutionary selection pressure on enzyme turnover numbers[5,15,17]. We confirmed this important role of flux by using fluxes based on experimental metabolic flux analysis (MFA) data instead of in silico fluxes, leading to very similar model performances (Supplementary Figure 5, see Methods). Likewise, the generalist feature is an important contributor in both models. Structural features are of consistent importance in both models,

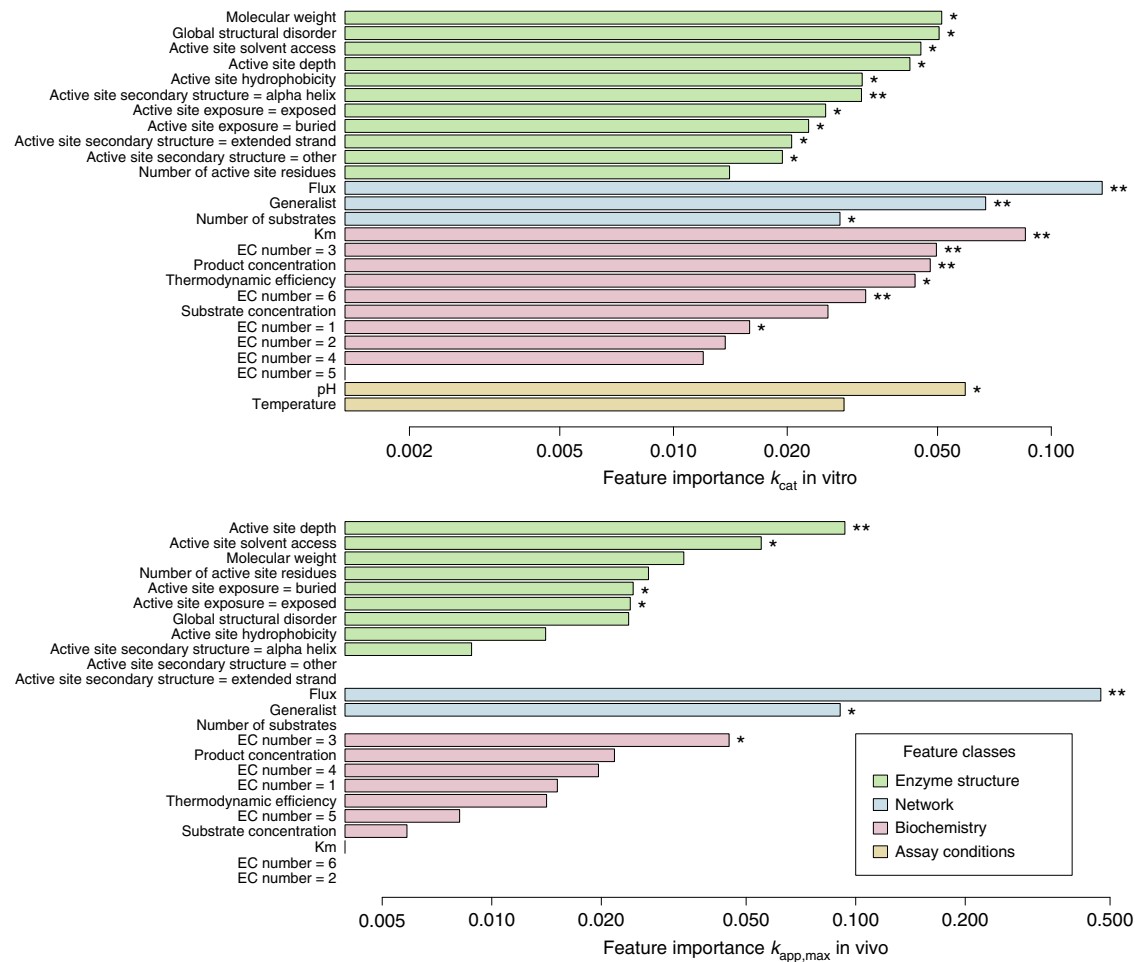

**Fig. 3** Feature importance in random forest models for in vivo and in vitro turnover numbers. The relative importance as measured by the average decrease in out-of-bag mean squared error (MSE) across trees that results from randomly permuting a given feature (scaled by the standard deviation) is shown. Missing bars indicate permutation importance smaller or equal to zero. The statistical significance of feature importance was evaluated using a permutation test based on 500 permutations of the response variable per model; *$p$-value < 0.05, **$p$-value < 0.005. Spearman rank correlation between the importance estimates of the two models is 0.47 ($p$ < 0.021, $n$ = 24, $S$ = 1214, see Methods), ignoring assay-related features that are not used in the model for $k_{app,max}$

with active site depth, active site solvent accessibility, and active site exposure showing significant contributions in both models. Interestingly, enzyme $K_m$ is a very important feature in the $k_{cat}$ in vitro model, but yields no predictive advantage in the model for $k_{app,max}$. This effect might be due to the original $k_{app,max}$ estimation being biased with regard to enzyme saturation.

**Machine learning models improve proteome predictions**. A major obstacle in the utilization of GEMs of protein investment is the requirement of thousands of direction-specific enzyme turnover rate constants (over 3000 in $i$ML1515), whereas both in vitro and in vivo data sets are limited to a few hundred of these measurements (497 and 234, respectively, covering 412 and 234 reactions, respectively; Supplementary Figure 2).

The high cross-validated accuracy of the ML models for $k_{app,max}$ (Fig. 2) suggests that these statistical models could be utilized to predict the $k_{app,max}$ of metabolic processes on a genome scale to improve the predictive accuracy of mechanistic GEMs. To achieve this goal, we created an ensemble model for $k_{app,max}$ that combines predictions across three diverse ML models: the linear elastic net, the decision-tree-based random forest model, and the

complex neural network model (see Methods and Supplementary Table 2 for details). The linear elastic net is expected to exhibit low variance at the cost of higher bias, whereas the two more complex algorithms, the random forest and the neural network, are more prone to overfitting on the relatively small dataset[36]. We confirmed this behaviour by computing learning curves (Supplementary Figure 4). Model training and genome-scale predictions are limited by the number of feature observations available for each reaction, suggesting that imputation of missing feature observations may lead to more accurate ML models (Supplementary Figure 2). For each of the three ML algorithms, we thus trained four versions: one without imputation, one with imputation of the training set, one with imputation of only the features predictions are based on, and one where all observations are imputed (see Methods for details). In cases where observations contained missing values that were not imputed, the median across all successful predictions was used. The diversity of these ML models is reflected in the modest correlation of their predictions (average $R^2$ between predictions is 0.27 for $k_{app,max}$ and 0.08 for $k_{cat}$ in vitro) suggesting that an ensemble approach may improve ML model accuracy. We thus used the average

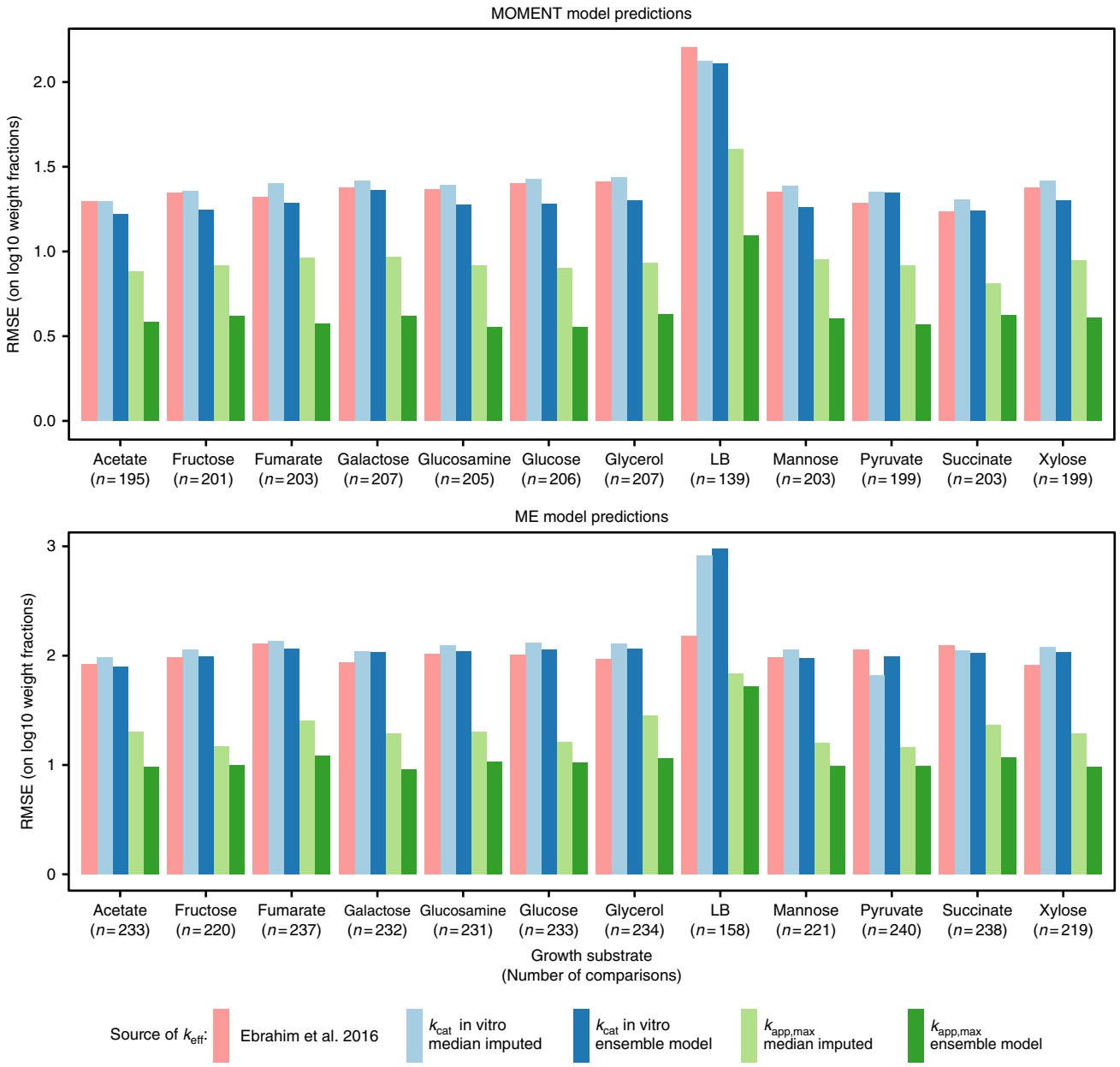

**Fig. 4** Performance of vectors of catalytic turnover numbers in predicting quantitative proteome data. Performance for two different genome-scale metabolic modelling frameworks, MOMENT and the ME model, are shown. Model predictions are compared to quantitative proteomics data in Schmidt et al.[37] through the root mean squared error (RMSE) for metabolic proteome fractions on log10 scale. Comparisons use proteins that are both found in proteomics data and are expressed in the model predictions. To allow comparison of different parameterization strategies, the intersection of the sets of comparable proteins is used in each condition-model combination resulting in the number of comparisons n. The performance of the two modelling frameworks, MOMENT and ME, is thus not comparable, as different sets of proteins are underlying the error calculations. See Methods and Supplementary Figure 8 for details

prediction across these twelve models as the final ensemble model. Experimental data on $k_{app,max}$ and $k_{cat}$ in vitro was then extrapolated to the genome scale using the respective ensemble model.

Enzyme catalytic turnover numbers strongly affect the proteomic cost of reaction fluxes. The predictive performance of GEMs for quantitative proteome allocation is thus expected to be sensitive to the set of effective turnover rates. We used two different GEM modelling frameworks, metabolic modelling with enzyme kinetics (MOMENT)[5] and a GEM of metabolism

and gene expression (ME model)[8,9], to predict quantitative proteomics data[37] and to compare predictive performance across different genome-scale parameterization strategies: known in vitro $k_{cat}$ with missing values simply replaced by the median of known values (median-imputed), in vitro $k_{cat}$ extrapolated with the ensemble ML model, median-imputed $k_{app,max}$, and $k_{app,max}$ extrapolated with the $k_{app,max}$ ML ensemble model (Fig. 4). Furthermore, we also included a parameterization with a fit of selected $k_{eff}$ parameters to proteomics data that was conducted earlier to study the

regularity of $k_{eff}$s[12]. The major difference between the MOMENT algorithm and the ME model lies in the fact that ME models explicitly model the details of gene expression machinery and co-factor synthesis, resulting in a more realistic representation of enzyme complex stoichiometry and growth rate-dependent gene product dilution.

We find that predictive capability of both MOMENT and the ME model is higher for $k_{app,max}$-based parameter sets than for those based on $k_{cat}$ in vitro, where the prediction error is on average 43% lower in MOMENT and also 43% lower in the ME model. The ensemble ML model further improves the predictive performance of $k_{app,max}$-based GEMs consistently across growth conditions and mechanistic modelling techniques, with an average reduction in root mean squared error (RMSE) of 34% and 20% for MOMENT and the ME model, respectively (Fig. 4). As expected from the high cross-validation errors for the ML models of $k_{cat}$ in vitro (Fig. 2), the gain in performance that originates from the ensemble ML model for $k_{cat}$ in vitro is much lower than that of the $k_{app,max}$ ML model, with an average reduction in RMSE of 7% for MOMENT and 1% for ME models (Fig. 4).

## Discussion

The diversity of biochemical reactions renders genome-scale experimental characterization of enzyme kinetics a task of prohibitive complexity. We show that ML models of enzyme structure, network context, and biochemistry can be utilized for the in silico prediction of catalytic turnover numbers, particularly in the case of in vivo estimates of apparent enzyme turnover, $k_{app,max}$. How does the well-performing ML model of $k_{app,max}$ arrive at its predictions? In agreement with the hypothesis of differential selection pressure on catalytic turnover numbers that is determined by enzyme utilization[5,15,17], the model predicts higher turnover numbers for enzymes that carry high flux across diverse growth conditions (Supplementary Figure 7). This effect is likewise found in the model for $k_{cat}$ in vitro (Supplementary Figure 6) and flux is also the most important feature in the in vitro model (Fig. 3). Furthermore, the ML model for $k_{app,max}$ predicts a decline of enzyme catalytic turnover rates with depth of the active site (Supplementary Figure 7), a result consistent with diffusion-limited theory of catalysis in enzymes with buried active sites, which predicts a decrease with tunnel depth[38]. Similarly, a negative impact of solvent accessibility on enzyme turnover rates is inferred in the $k_{app,max}$ model. This result is in agreement with multiple observations of the importance of selective barriers that prevent water access of the active site for enzyme function (reviewed by Gora et al.[39]). The tendency of enzyme components to catalyze multiple reactions (the generalist property) was identified as a major contributor to predictions in models of $k_{app,max}$ and $k_{cat}$ in vitro, where multifunctional components tend to decrease catalytic turnover rates (Supplementary Figure 6 and 7). This finding agrees with reports that in vitro $k_{cat}$s of specialist enzymes are higher than that of other enzymes[28] and the trade-off between multi-functionality and catalytic activity observed in directed evolution experiments[40]. The mechanism of the reaction catalyzed by a given enzyme, coded by the first digit of its EC number, was previously found to be correlated with in vitro $k_{cat}$[15]; interestingly, EC numbers only play a minor role in the predictions of models for both $k_{cat}$ in vitro and $k_{app,max}$ (Fig. 3). This minor role of catalytic mechanism in comparison to evolutionary factors is supported by a recent analysis of in vitro $k_{cat}$s in the context of spontaneous reaction rates[41]. Interestingly, the Michaelis constant ($K_m$) is a very important feature in the model for $k_{cat}$ in vitro, but plays no significant role in the $k_{app,max}$ model. One possible reason is that $k_{app,max}$, as an estimator of $k_{cat}$, is expected to be biased in

terms of $K_m$, where the bias acts in the opposite direction from the effect estimated for $k_{cat}$ in vitro.

In vitro $k_{cat}$ and $k_{app,max}$ originate from disparate sources. Thus, the agreement between the ML models for in vitro $k_{cat}$ and $k_{app,max}$ in terms of feature importance hierarchy (Fig. 3) and learned feature-output interaction of the most important features (Supplementary Figures 6 and 7) indicates that the ML approach identified meaningful determinants of catalytic turnover rates. Furthermore, the training data sets that were used to train the two models showed only a small overlap (39% of reactions with known $k_{app,max}$ have $k_{cat}$ in vitro associated, 22% of reactions with known $k_{cat}$ in vitro have $k_{app,max}$ associated, see Supplementary Figure 2), supporting the notion that meaningful global trends were identified. Nevertheless, the low predictive performance of the $k_{cat}$ in vitro model suggests that the model structure of this model should be interpreted with care.

Prediction accuracy for $k_{app,max}$ was consistently found to be significantly higher than that of $k_{cat}$ in vitro (Fig. 2). One possible explanation for this effect is the high level of noise in in vitro data: a global comparison of in vitro $k_{cat}$ data from the BRENDA database[15] found considerable discrepancies between $k_{cat}$s of the same reaction that were measured by different laboratories. These discrepancies are possibly due to technical difficulties of in vitro enzyme assays, e.g., in vitro–in vivo effects[15], erroneous database entries[15], and posttranslational modifications[42,43]. In contrast, $k_{app,max}$ is derived globally from few proteomics datasets, thus considerably decreasing the number of experimental sources and increasing comparability across the proteome. Another explanation for the superior performance of ML models for $k_{app,max}$ might lie in the fact that in silico fluxes were used to estimate $k_{app,max}$[14], and we likewise used in silico fluxes in this study. We show that this is not the case, as using fluxes based on MFA data in our framework does not decrease model performance (Supplementary Figure 5).

We utilized genome-scale metabolic models that account for the proteome costs of metabolic fluxes to test the ability of naively imputed and ML model-predicted vectors of $k_{app,max}$ and $k_{cat}$ in vitro to explain measured proteome investment across different carbon sources. Although the vector of effective enzyme turnover rates is a condition-dependent property because it depends on substrate concentrations and regulation, using the upper limit on effective turnover rates in the form of $k_{cat}$ in vitro or $k_{app,max}$— where a $k_{cat}$ in vitro is theoretically an upper limit on $k_{app,max}$—is expected to provide a reasonable default parameterization of these constraint-based models. We find that the traditional practice of using $k_{cat}$ in vitro[5,6] is consistently outperformed by parameterization using $k_{app,max}$ (Fig. 4). This finding might be due to the high noise level in $k_{cat}$ in vitro data discussed above, and important in vivo effects that are not captured by in vitro assays, like backwards flux in thermodynamically unfavourable reactions and regulatory effects[14]. Perhaps more importantly, $k_{app,max}$ estimation included the Schmidt et al.[37] dataset, and performance comparisons with $k_{cat}$ in vitro might thus be optimistically biased in favour of $k_{app,max}$. We verify the superior performance of $k_{app,max}$ on an independent dataset for chemostat growth on glucose minimal medium[44] and again find a clear advantage of using $k_{app,max}$ compared to $k_{cat}$ in vitro, with an average reduction in RMSE of 51% for MOMENT and 46% for the ME model. Surprisingly, the set of $k_{eff}$s that was obtained by Ebrahim et al.[12] yielded a performance comparable to the in vitro $k_{cat}$ parameterizations, even though it was obtained as a fit to the proteomics data set we are using as validation. This behaviour could be explained by the fact that Ebrahim et al. aimed to study biological regularities, and thus only used fitted $k_{eff}$ parameters that are invariable across conditions, and focused on highly expressed proteins in their optimization procedure.

Did the statistical models of enzyme turnover numbers learn to make meaningful predictions? The ensemble model for $k_{app,max}$ outperforms all other parameter sets across all growth conditions for both MOMENT and ME model algorithms in terms of predictive performance for quantitative proteome data. In comparison to simple median-imputation, the ML model of $k_{app,max}$ reduces the RMSE by 34% for MOMENT and by 20% for the ME model. This result indicates that the ensemble ML model of $k_{app,max}$ has identified meaningful features that allow for an improvement of the genome-scale estimation of catalytic turnover rates. As expected from the higher cross-validated performance that was estimated for ML models of $k_{app,max}$ (Fig. 2), the improvement in performance that is achieved by the ensemble models compared to naive imputation is higher for $k_{app,max}$ than it is for $k_{cat}$ in vitro (Fig. 4).

A major limitation of statistical modelling of catalytic turnover numbers is the comparatively small size of the datasets for $k_{cat}$ in vitro and $k_{app,max}$ (497 and 234 observations in this study, respectively). The most promising output, $k_{app,max}$, is currently limited to unique homomers—i.e., the enzyme subunit is only used in one reaction—and to reactions that have proteomics data and flux predictions available. Our current ML model of $k_{app,max}$ is thus likely biased toward unique homomeric enzymes. Careful extension of the $k_{app,max}$ protocol to non-unique and heteromeric proteins, flux estimation of non-essential reactions, or extension of the scope of expression data via ribosome profiling could be used to further improve genome-scale estimation; learning curves for the complex random forest model confirm that additional data is likely to increase model performance (Supplementary Figure 4). Furthermore, data for both $k_{cat}$ in vitro and $k_{app,max}$ on membrane proteins is scarce. Membrane components are thus a promising target for future statistical and experimental analysis, as they are responsible for growth-critical tasks like transport and oxidative phosphorylation. Finally, given the condition-dependent nature of $k_{eff}$, context-specific statistical models for $k_{eff}$ are a promising avenue to further improve the predictive performance of mechanistic metabolic models.

The proteomic costs of metabolic fluxes are of significant importance for our understanding of the cell as a system, but experimental procedures for determining enzyme turnover numbers are not suitable for genome-scale applications. The ML models we present give extensive insight into the global determinants of enzyme turnover numbers and improve our understanding of the kinetome—and thus the quantitative proteome—of E. coli.

## Methods

**Calculating flux states using parsimonious FBA**. We calculate parsimonious FBA[27] solutions for iML1515, a GEM of E. coli K-12 MG1655[26]. Linear programming problems were constructed using the R[45] packages sybil[46] and sybilccFBA[47], and problems were solved using IBM CPLEX version 12.7. A single iteration of this sampling algorithm proceeds as follows: Oxygen uptake was allowed with probability 1/2, and the environment always contained at least one randomly chosen source of each carbon, nitrogen, sulfur, and phosphate. A number of additional sources per element were drawn from a binomial of size 2 with success probability 1/2. Carbon uptake rates were normalized to the number of carbon atoms in the selected substrates. This process was repeated until a growth sustaining environment was found and the flux distribution recorded, concluding the iteration. Using this algorithm, we simulated 10,000 environments, and averaged these flux distributions across environments to arrive at the flux feature.

**Calculating MFA-constrained flux states**. As an alternative to the flux sampling using parsimonious FBA, experimental data on metabolic flux obtained from metabolic flux analysis (MFA) was utilized (presented in Supplementary Figure 5). Reaction fluxes estimated from MFA were obtained for eight growth conditions for E. coli[48]. FBA using the E. coli metabolic network reconstruction iML1515[26] was then used to identify a steady-state flux distribution ($\mathbf{v}_{FBA}$) as close to the MFA-

estimated values ($\mathbf{v}_{data}$) as possible using a quadratic programming (QP) problem:

$$\text{Min} \sum_i \left( v_{FBA,i} - v_{data,i} \right)^2 \text{s.t.} \tag{1}$$

$$\mathbf{S}\mathbf{v}_{FBA} = 0$$

$$v_{lb,i} < v_{FBA,i} < v_{ub,i}$$

For each condition, the Pearson correlation between MFA-estimated and FBA-calculated fluxes was greater than 0.99, indicating general concordance between the model used to estimate the MFA fluxes and iML1515.

Measured fluxes were then constrained to their QP-optimized values, and FBA was once again run with an ATP maximization objective (termed the ATP maintenance reaction or ATPM)[49] by solving a linear programming (LP) problem:

$$\text{Max } v_{ATPM} \text{s.t.} \tag{2}$$

$$\mathbf{S}\mathbf{v}_{FBA} = 0$$

$$v_{lb,i}^* < v_{FBA,i}^* < v_{ub,i}^*$$

where $\mathbf{v}_{lb}{}^*$ and $\mathbf{v}_{ub}{}^*$ are the standard flux bounds augmented with the QP-optimized values from Eq. (1).

Finally, the objective ATP production reaction was set to its calculated optimal value, and the total flux was minimized subject to all previous constraints as a parsimony objective based on the idea that the cell generally will not carry large amounts of unnecessary flux due to the cost of sustaining the required enzyme levels[50].

$$\text{Min } \|\mathbf{v}_{FBA}\|_2 \text{ s.t.} \tag{3}$$

$$\mathbf{S}\mathbf{v}_{FBA} = 0$$

$$v_{lb,i}^{\#} < v_{FBA,i}^{\#} < v_{ub,i}^{\#}$$

where $\mathbf{v}_{lb}{}^{\#}$ and $\mathbf{v}_{ub}{}^{\#}$ are the same flux constraints used in the problem defined in Eq. (2) but now augmented with a constraint on the optimal value of $v_{ATPM}$ identified in Eq. (2).

The final flux solutions show good agreement with MFA-estimated flux states, including measured growth rates, while maximizing ATP production and maintaining parsimony as secondary objectives. The average of the final flux solutions in the eight growth conditions was used as the flux feature for the sensitivity analysis shown in Supplementary Figure 5. Problems were set up using the COBRA toolbox version 2.0 in Matlab 2016b and solved using Gurobi 8.0.1 solvers.

**Generalist property**. Based on the GPR relations provided by iML1515, we use the maximum number of times the gene products catalyzing a given reaction are utilized in other reactions to quantify the generalist feature. The number of substrates for a given reaction were extracted from the stoichiometric matrix of iML1515, excluding water and protons.

**Protein sequence and structure property calculations**. To gather protein-specific features, global properties of catalytic enzymes and local properties of their active sites were calculated using the ssbio Python package[51]. First, model reactions in iML1515 were mapped to their protein sequences and 3D structures based on the stored GPR rules. This was done utilizing the UniProt mapping service, allowing gene locus IDs (e.g., b0008) to be mapped to their corresponding UniProt protein sequence entries (e.g., P0A870) and annotated sequence features[52]. Next, UniProt identifiers were mapped to structures in both the Protein Data Bank[29] and homology models from the I-TASSER modelling pipeline[31]. These structures were then scored and ranked[53] to select a single representative structure based on resolution and sequence coverage parameters. For the cases in which only PDB structures were available, the PDBe best structures API was queried for the top scoring structure. If no more than 10% of the termini were missing along with no insertions and only point mutations within the core of the sequence, the structure was set as representative. Otherwise, a homology model was selected by sequence identity percentage or queued for modelling[53]. It is important to note that the structure selection protocol results in a final structure that is monomeric, and thus parameters which may be impacted by quaternary complex formation are not

currently considered. This is a limitation in both experimental data and modelling methods, as complex structures remain a difficult prediction to make. Furthermore, for global and local calculations (described below), all non-protein molecules (i.e., water molecules, prosthetic groups) were stripped before calculating the described feature. Out of the 1515 proteins, 729 experimental protein structures and 784 homology models were used in property calculations. Finally, we added annotated active site locations from the Catalytic Site Atlas SQL database[32] for any matching PDB ID in the analysis.

Global protein properties were classified as properties that were derived from the entire protein sequence or structure (e.g., percent disordered residues), and local properties were those that described an annotated catalytic site (e.g., average active site depth from the surface). From the protein sequence, global properties were calculated using the EMBOSS pepstats package[54] and the Biopython ProtParam module[55]. Local properties for secondary structure and solvent accessibilities were predicted from sequence using the SCRATCH suite of tools[56] and additionally calculated from set representative structures using DSSP[57] and MSMS[58]. Predicted hydrophobicities of amino acids were calculated using the Kyte-Doolittle scale for hydrophobicity with a sliding window of seven amino acids[59]. For a full list of obtained properties, see Supplementary Table 2.

**Biochemical features.** Reaction EC numbers were obtained from the Bigg database[60], and extended with additional EC number data from KEGG[61] and MetanetX[62] where available.

To estimate reaction Gibbs energies, metabolite data for eight growth conditions for *E. coli* was obtained from literature[48]. Reaction equilibrium constants ($K_{eq}$s) were estimated using the latest group contribution method[63]. Then, a thermodynamic FBA problem[64] was solved constraining only high flux reactions (>0.1 mmol/gDW/h), subject to uncertainty. Once a feasible set of fluxes, metabolite concentrations (x), and $K_{eq}$s was identified, convex sampling was used to obtain a distribution of x and $K_{eq}$ values that accounts for measurement gaps and uncertainty. These sampled x and $K_{eq}$ values were used to calculate the reaction Gibbs energies using the definition:

$$\Delta G = -\text{RTlog}\left(K_{eq}\right) + \log(Q)$$
$$Q = \prod_i x_i^{S_i}$$

where Q is the reaction quotient defined as the product of the metabolite concentrations (or activities) to the power of their stoichiometric coefficient in the reaction (S). The thermodynamic efficiency parameter $\eta_{rev}$ used in this study was then calculated from this $\Delta G$ using its definition[65]:

$$\eta_{rev} = 1 - \exp(\Delta G/RT) = 1 - Q/K_{eq}$$

Note that this expression is bounded between 0 and 1 for reactions in the forward direction ($\eta_{rev}$ is 0 at equilibrium and 1 at perfect forward efficiency). For consistency, we considered each reaction as the forward direction stoichiometry for this calculation. Average $\eta_{rev}$ across the eight growth conditions was used as model input feature.

Michaelis constants ($K_m$s) were extracted from the BRENDA[33] and the Uniprot[52] resource and manually curated. When multiple values exist for the same constant, in vivo-like conditions, recency of the study, and agreement among values were used as criteria to select the best value.

The average metabolite concentrations across the eight growth conditions mentioned above[48] were used as features on substrate and product concentrations.

**Summarizing data across genes.** We summarized all features and outputs to the reaction level as given in the metabolic representation of the *E. coli* metabolic network iML1515. In the case of structural features, which were obtained at the gene-level, we used the GPR relations provided by the model to summarize features. Details are listed in Supplementary Table 1.

**Linearization.** Features and outputs were transformed to favour linear relationships between features and outputs. Flux, enzyme molecular weight, $K_m$, metabolite concentrations, $k_{cat}$ in vitro, and $k_{app,max}$ were log-transformed. The reciprocal of temperature was used as suggested by the Arrhenius relationship.

**Imputation.** The set of features does not contain data on all features for all reactions in iML1515 (See Supplementary Figure 2). To allow GEM predictions, we utilize different imputation strategies: imputation of labelled data, i.e., data that has outputs associated, only, imputation of the unlabelled data only, imputation of both labelled and unlabelled data, and no imputation. Missing observations were imputed using predictive mean matching for continuous data, logistic regression for binary data, and polytomous regression for categorical data of more than two categories (see Supplementary Table 1 for details). This procedure was implemented using the mice package in the R environment[45,66]. Output data was not used for imputation to prevent optimistic bias in error estimates.

**Data on $k_{cat}$ in vitro.** We extracted in vitro $k_{cat}$ values for enzymes occurring in the *E.coli* K-12 MG1655 iML1515 model from the BRENDA[33], Sabio[34], and Metacyc[35] databases. A total of 6812 $k_{cat}$ values were downloaded based on EC numbers. We removed redundant data points that originated from the same experiment in the same publication across databases. When deleting redundant data, we gave preference to the BRENDA and the Metacyc database, in that order. Next, we removed all data explicitly referring to mutated enzymes.

A central problem in using data from these three databases is that many $k_{cat}$ values were measured in the presence of unnatural substrates that are unlikely to occur in physiological conditions. We use the iML1515 model as a resource for naturally occurring metabolic reactions. To use this list as a filter, we mapped reactions from our curated datasets to model reactions. This reaction mapping was implemented using the synonym lists of substrates provided by the MetRxn resource[67]. Six hundred and sixty four database entries did not contain complete reaction formulas, and we mapped those based on EC numbers and substrate information. We manually checked all entries in the Metacyc dataset with the keyword 'inhibitor' in the experimental notes, and omitted data that was measured in the presence of inhibitors. Finally, in cases where multiple literature sources were available, we manually selected sources giving preference to in vivo-like conditions, recency of the study, and agreement among values, making additional use of data in the Uniprot Resource[52]. In the end, we are left with 497 useable $k_{cat}$ in vitro values that cover 412 metabolic reactions.

**Cross validation and hyperparameter tuning.** Statistical models of turnover rates were trained using the caret package[68] and, in the case of neural networks, the h2o package[69]. Model hyperparameters were optimized by choosing the set that minimizes cross-validated RMSE in five times repeated (One repetition in the case of neural networks) 5-fold cross-validation. In the case of neural networks, hyperparameters were optimized using 3000 iterations of random discrete search and 5-fold cross-validation. Details on implementation and hyperparameter ranges are given in Supplementary Table 2.

**Mechanistic model prediction of protein abundances.** In order to validate the ability of different vectors of catalytic turnover rates to explain quantitative protein data, proteome allocation was predicted using the MOMENT algorithm. We calculate MOMENT solutions for iML1515 using turnover rates obtained from the respective data source or ML model. In the case of membrane proteins, which were not in the scope of the ML model, a default value of $65\,s^{-1}$ was used. Linear programming problems were constructed using the R[45] packages sybil[46] and sybilccFBA[47], and problems were solved using IBM CPLEX version 12.7. Enzyme molecular weights were calculated based on the *E. coli* K-12 MG1655 protein sequences (NCBI Reference Sequence NC_000913.3), and the total weight of the metabolic proteome was set to 0.32 $g_{protein}/g_{DW}$ in accordance with the *E. coli* metabolic protein fraction across diverse growth conditions[5,44]. Aerobic growth on each substrate in Schmidt et al.[37] was modeled by setting the lower bound corresponding to the uptake of the substrate and oxygen to $-1000$ mmol gDW$^{-1}$ h$^{-1}$, effectively leaving uptake rates unconstrained.

In addition to MOMENT, a GEM of metabolism and gene expression (ME model)[8,9] was applied to validate the predicted enzyme turnover rates. For these simulations the iJL1678b ME-model of *E. coli* K-12 MG1655 was used[70]. Like in the MOMENT predictions, a default value of $65\,s^{-1}$ was used for the $k_{eff}$s of membrane proteins, and aerobic growth on each substrate in Schmidt et al.[37] was modeled by setting the lower bound corresponding to the uptake of the substrate and oxygen to $-1000$ mmol gDW$^{-1}$ h$^{-1}$, effectively leaving uptake unconstrained. The $k_{eff}$s of all processes in iJL1678b-ME that fell outside the scope of iML1515 were also set to $65\,s^{-1}$. The model was optimized using a bisection algorithm and the qMINOS solver, a solver capable of performing linear optimization in quad-precision[71,72], to find the maximum feasible growth rate within a tolerance of $10^{-14}$. The unmodeled protein fraction, a parameter to account for expressed proteins that are either outside the scope of the model or underutilized in the model, was set to 0. Further, mRNA degradation processes were excluded from the ME-model for these simulations to prevent high ATP loads at low growth rates.

Genes that are subunits in membrane localized enzyme complexes and genes involved in protein expression processes were out of the scope of the $k_{app,max}$ and $k_{cat}$ in vitro prediction approaches. Thus these genes were not considered when comparing predicted and measured protein abundances (Fig. 4). In silico predictions that had an abundance greater than zero were matched to experimental protein abundances if the latter contained more than 0 copies/cell. Weight fractions of the metabolic proteome were estimated by normalizing by the sum of masses for in silico predictions and experimental data, respectively.

**Statistics.** The statistical significance of Spearman's $\rho$ correlations was tested using the AS 89 algorithm[73] as implemented in the cor.test() function of the R environment[45]. Permutation tests for feature importance in the random forest models were conducted using the R package rfPermute using 500 permutations of the respective response variable per model.

**Code availability.** R code for model training and analysis, and Python code for ME modelling are available from the authors upon request.

## Data availability

The vectors of turnover numbers predicted by our ensemble models alongside the experimental benchmarks used in Fig. 4 are available as Supplementary Data 1. A Reporting Summary for this Article is available as a Supplementary Information file.

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

## Acknowledgements

This work was supported by the Novo Nordisk Foundation Grant Number NNF10CC1016517. M.J.L. was supported by the German Research Foundation (CRC 1310). The authors would like to thank Marc Abrams for proofreading the manuscript.

## Author contributions

D.H., M.J.L., and B.O.P. designed research. Y.H. and Z.H. extracted and filtered in vitro $k_{cat}$ data under supervision of D.H. N.M. and D.H. compiled protein structure data. D.C. Z. contributed features from MFA data, thermodynamic efficiencies, and $K_m$, and further curated in vitro $k_{cat}$ data. D.H. conducted machine learning modelling and analysis. C.J.L. conducted ME modelling. D.H. and A.A.D. conducted MOMENT modelling. D.H. and B.O.P. wrote the paper.

## Additional information

**Competing interests:** The authors declare no competing interests.

