## [Peer Review File · Nature Communications]

Reviewers' comments:

Reviewer #1 (Remarks to the Author):

The question addressed by the article is related to a classic question: what are the features of enzymes that determine (allow to predict) their "efficiency" (k_{cat} or $k_{app\ max}$).

In view of the heterogeneity of the enzyme features, it is attractive to tackle the question with Machine Learning (ML) techniques.

Recent advances in the absolute quantification of proteins are a major renewal of the question and therefore provide an interesting opportunity to revisit it. Indeed, as recently highlighted in an article of Milo, and recalled in the article, the values of k_{cat} measured in vitro, indexed in databases like Brenda, are characterized by a very large variance. This large variance is an important limitation for applying ML approaches to these data. The article confirms this, given the very poor quality of predictions made on the basis of the k_{cat} values training set.

The authors took a very different line. Indeed, they implicitly assume that the proteomic data set remains very limited. Indeed, they only consider that 123 values of $k_{app\ max}$ are available for *E. coli*. This number seems especially low in view of the available protein data set (as the one recently published Schmidt A. et al.).

In view of the importance of the training set size in ML, I do not understand why the authors did not use this data to have access to a better cover of all enzyme types. The "validation section" specifically disturbed me in this context. Indeed, authors validated the quality of the prediction with Schmidt data proteomic set. If the quality of the data is questionable, then why use it to validate the prediction? and if not, why did they not use the data to significantly increase the size of the training set?

The conclusion of all this is that I do not know what the authors are trying to do: if the objective is to predict $k_{app\ max}$, why don't authors use available data to significantly increase the learning set? If the objective of the paper is to investigate whether $k_{app\ max}$ can be deduced on the basis of enzyme features, why don't authors use available data to significantly increase the learning set?

Furthermore, it seems that ongoing advances in quantitative proteomics suggest that the issue of estimating $k_{app\ max}$ for ME-type prediction is more related to flux measurement/estimation than to the protein quantification.

Indeed, a current limit in protein quantification is associated with proteins with very low abundance (typically less than 10 copies per cell), but these enzymes actually have almost no cost and therefore do not really have an impact on ME type prediction. Finally, it is necessary to mention in this context, recent progress in ribosome profiling which, in growth regime, can bypass some limitations of MS type methods...

Reviewer #2 (Remarks to the Author):

The authors present a data-driven approach for estimating optimal enzyme turnover rates which are important for understanding cellular metabolism in the context of enhancing the accuracy of genome-scale models. They also identify a diverse set of features that are predictive for both in vivo/in vitro enzyme turnover rates, revealing novel protein structural correlates of catalytic turnover. The overall research is of high quality and high novelty. I do not have major concerns. A few minor suggestions are following.

1. Figure 1 showed two assay conditions as features (pH and Temperature). However, other cultivation conditions (such as bioreactor modes, mixing condition, addition of rich nutrients, oxygen, etc...) may strongly influence cell growth rate and in vivo fluxes. Authors might consider more features from bioprocess factors to improve model predictions. They should discuss previous machine learning reports that used different features to predict cell performance or fluxes (Biotechnology and Bioengineering. 2011. 108(4): 893–901; Plos Computational Biology. 2016. 12(4):e1004838; ChemBioEng Reviews. 2016, 3, No. 2, 1–11).

2. In Figure 2, cross-validated machine learning was based on R square. The R square for k-cat was up to 0.75 (relatively poor prediction quality). How many runs were used for each method to calculate cross-validated R²? If RMSE was used to compare performance, would authors have same conclusions about machine learning models?

3. It would be of great value to see the performance of the models with other E. coli genome-scale metabolic network reconstructions. Particularly, the latest model, iML1515 (Nature Biotechnology volume 35, pages 904–908 (2017)).

4. The authors do not present a learning curve (graph that compares the training and cross validation error over a varying number of training instances) to see how the performance of these models depend on the size of the datasets (which are quite small). It is critical to understand how the accuracies would vary with the amount of data available.

5. A good number of the features involved a lot of computations and could potential be sources of error. The robustness of the machine learning models to these errors should be quantified.

Reviewer #3 (Remarks to the Author):

This work describes a machine learning based workflow to estimate enzyme catalytic activity. The authors have highlighted the key mechanistic features to train different machine learning models and have also applied the results from this study to parameterize two different protein cost models (i.e. MOMENT and ME model). Their results show improved quantitative proteome predictions by using the parameterized apparent turnover number (i.e. $k_{app,max}$). However, the authors did not clearly state the source of their training dataset, and their model features, model testing metric, and the comparisons with existing procedures are not thoroughly convincing as explained below.

Major Concerns

- MOMENT requires $k_{app,max}$ values to estimate fluxes. The authors mention that the $k_{app,max}$ values used in the training dataset for the Machine learning (ML) models were calculated using MOMENT. It is not clear which fluxes/datasets were used to estimate the training dataset. The authors have also mentioned that they used MOMENT to test their ML model predictions and observed improved predictions. Thus, it appears that the testing was better than training while using data from the same source. Could the authors comment on this?
- The authors have used the average of flux distributions, under varying nutrient conditions estimated using MOMENT, as a feature during model training. However, this feature does not represent a meaningful flux distribution in *E. coli*. The reviewer would instead suggest that fluxes estimated using ¹³C-MFA data can be used as a feature in the training data.
- The authors have reported that their k_{cat} in vitro predictions had an average R² value of 0.22 compared to a value of 0.75 for $k_{app,max}$ predictions. The reviewer is wondering if the authors

could improve the kcat in vitro further by using more features for training. It is also not convincing that the features are “important” due to low prediction fidelity of the kcat in vitro model. It is highly probable that the “important” features would vary when the accuracy of machine learning model of kcat in vitro is improved.

- A few important features are missing in the feature selection section of the ML models. kapp,max varies as a function of metabolite concentrations, ΔG , and km based on the study by Noor et al. [1]. However, the authors did not use these parameters as features for their machine learning models. There are few studies [2, 3] that predict km using machine learning approaches. Thus, the features that correlate with km can also be included to contribute towards kapp,max predictions. In addition, a recently published metabolomics dataset [4] can also be included in the feature set for metabolite concentrations.
- The accuracy of all the machine learning model are quantified using cross-validation R2 values in Figure 2. These methods exhibit good accuracies (>75% in cross validation), but it is still possible that they may not perform well in the predictions (see SVM model in study [3]). An unbiased set of data should be used to clearly quantify the accuracy of prediction in this workflow (i.e. partition the dataset into train, validate, and test datasets).
- The authors did not explain why the kapp,max-based model has a better prediction than the model using the keff dataset from Ebrahim et al. [5]. The set of keff from Ebrahim et al. was fitted by minimizing the difference between model predicted proteome and experimentally measured proteome using the data from Schmidt et al. [6]. Theoretically, this set of keff should give the best prediction (because the objective is similar to minimization of RMSE in Figure 4). It is surprising that kapp,max-based model can improve the already minimized value obtained from the constraint optimization problem by Ebrahim et al. [5].

Minor Concerns

- As mentioned by the author in line 270-271, protein cost model by using parameters trained in this machine learning model can only provide an upper limit because effective enzyme turnover rate is condition-dependent. Is it possible to include more machine learning features (based on each condition) to obtain a more accurate model for each condition?
- Line 197 to 215: the last section seems redundant to the reader because the authors have already shown that similar feature importance exists between ML models of kapp,max and kcat. It is expected that an ensemble of ML models will have consistent important features. Can the authors comment on this?
- In page 9 line 212, the authors should replace “...that only show only small...” with “...that show only small ...”.

References:

1. Noor, E., et al., A note on the kinetics of enzyme action: A decomposition that highlights thermodynamic effects. *Febs Letters*, 2013. 587(17): p. 2772-2777.
2. Carbonell, P. and J.L. Faulon, Molecular signatures-based prediction of enzyme promiscuity. *Bioinformatics*, 2010. 26(16): p. 2012-9.
3. Mellor, J., et al., Semisupervised Gaussian Process for Automated Enzyme Search. *ACS Synth Biol*, 2016. 5(6): p. 518-28.
4. McCloskey, D., et al., RapidRIP quantifies the intracellular metabolome of 7 industrial strains of *E. coli*. *Metab Eng*, 2018. 47: p. 383-392.
5. Ebrahim, A., et al., Multi-omic data integration enables discovery of hidden biological regularities. *Nat Commun*, 2016. 7: p. 13091.
6. Schmidt, A., et al., The quantitative and condition-dependent *Escherichia coli* proteome. *Nat Biotechnol*, 2016. 34(1): p. 104-10.

Reviewers' comments with responses and actions taken:

Reviewer #1 (Remarks to the Author):

The question addressed by the article is related to a classic question: what are the features of enzymes that determine (allow to predict) their "efficiency" (k_{cat} or $k_{app\ max}$).

In view of the heterogeneity of the enzyme features, it is attractive to tackle the question with Machine Learning (ML) techniques.

Recent advances in the absolute quantification of proteins are a major renewal of the question and therefore provide an interesting opportunity to revisit it. Indeed, as recently highlighted in an article of Milo, and recalled in the article, the values of k_{cat} measured in vitro, indexed in databases like Brenda, are characterized by a very large variance. This large variance is an important limitation for applying ML approaches to these data. The article confirms this, given the very poor quality of predictions made on the basis of the k_{cat} values training set.

The authors took a very different line. Indeed, they implicitly assume that the proteomic data set remains very limited. Indeed, they only consider that 123 values of $k_{app\ max}$ are available for *E. coli*. This number seems especially low in view of the available protein data set (as the one recently published Schmidt A. et al.).

In view of the importance of the training set size in ML, I do not understand why the authors did not use this data to have access to a better cover of all enzyme types. The "validation section" specifically disturbed me in this context. Indeed, authors validated the quality of the prediction with Schmidt data proteomic set. If the quality of the data is questionable, then why use it to validate the prediction? and if not, why did they not use the data to significantly increase the size of the training set?

The conclusion of all this is that I do not know what the authors are trying to do: if the objective is to predict $k_{app\ max}$, why don't authors use available data to significantly increase the learning set? If the objective of the paper is to investigate whether $k_{app\ max}$ can be deduced on the basis of enzyme features, why don't authors use available data to significantly increase the learning set?

Response: Before we explain why the scope of $k_{app,max}$ is limited, we would like to point out that the model does not make use of only 123 values of $k_{app,max}$. We assume that the Reviewer is referring to the number of "complete observations" (l. 102 of the original manuscript). These 123 values of $k_{app,max}$ are those for which original data (i.e., without imputation) is available for all

features under consideration. The ensemble model used for prediction of proteomics data makes use of all 234 $k_{app,max}$ values provided in Davidi et al. (2016).

Now to the Reviewer's original question: why is the number of available $k_{app,max}$ values lower than the number enzymes that have their underlying gene products quantified in the Schmidt et al. proteomics data? Firstly, the calculation of $k_{app,max}$ requires a unique association between a gene-product and a catalyzed flux. Because of this requirement, Davidi et al. (2016) limited $k_{app,max}$ estimation to enzymes that are uniquely catalyzed by a single gene product.

Furthermore, calculation of $k_{app,max}$ requires *in silico* reaction flux to be greater zero in at least one of the available growth conditions. Thus, it is not the quality of the proteomics data, but these two limitations that preclude genome-scale estimation of $k_{app,max}$, limiting the analysis to the 234 values that we use to train our model. As we mention in the discussion, it would indeed be promising to explore an extension of the original $k_{app,max}$ protocol by Davidi et al. (2016) to increase its scope. That being said, we now added learning curves as Supplementary Figure 4, showing that, while there would likely be a small benefit to having a larger dataset, we are already in a region of diminishing returns, even for the complex random forest model.

Actions: We now give the number of complete observations before and after imputation in the Results section (lines 108-111). Learning curves (Supplementary Figure 2) now show the effect of training set size on predictive performance and we discuss the estimated effect of addition of new data accordingly (lines 356-358).

Furthermore, it seems that ongoing advances in quantitative proteomics suggest that the issue of estimating $k_{app,max}$ for ME-type prediction is more related to flux measurement/estimation than to the protein quantification.

Indeed, a current limit in protein quantification is associated with proteins with very low abundance (typically less than 10 copies per cell), but these enzymes actually have almost no cost and therefore do not really have an impact on ME type prediction. Finally, it is necessary to mention in this context, recent progress in ribosome profiling which, in growth regime, can bypass some limitations of MS type methods...

Response: As we explain above, we agree that flux estimation is the central problem that limits the scope of $k_{app,max}$ estimation in the original protocol of Davidi et al. (2016), in addition to the requirement of assigning gene products uniquely to reaction flux. Nevertheless, we think that—although they will not have a strong effect on growth rate prediction with ME models—low-abundance proteins should ultimately be parameterized correctly to allow, for example, predictions of differential expression. We thus now mention the possibility of using ribosome profiling data for extending the coverage of protein abundance quantification.

Actions: We extend our discussion of the limitations of the scope of $k_{app,max}$ by highlighting the importance of flux measurements for low-flux reactions, the possibility of using ribosome profiling data, and the results of our learning curve analysis (lines 354-358).

Reviewer #2 (Remarks to the Author):

The authors present a data-driven approach for estimating optimal enzyme turnover rates which are important for understanding cellular metabolism in the context of enhancing the accuracy of genome-scale models. They also identify a diverse set of features that are predictive for both in vivo/in vitro enzyme turnover rates, revealing novel protein structural correlates of catalytic turnover. The overall research is of high quality and high novelty. I do not have major concerns. A few minor suggestions are following.

1. Figure 1 showed two assay conditions as features (pH and Temperature). However, other cultivation conditions (such as bioreactor modes, mixing condition, addition of rich nutrients, oxygen, etc...) may strongly influence cell growth rate and in vivo fluxes. Authors might consider more features from bioprocess factors to improve model predictions. They should discuss previous machine learning reports that used different features to predict cell performance or fluxes (Biotechnology and Bioengineering. 2011. 108(4): 893–901; Plos Computational Biology. 2016. 12(4):e1004838; ChemBioEng Reviews. 2016, 3, No. 2, 1–11).

Response: We agree that *in vivo* conditions could affect the outcome of assays for k_{cat} *in vitro*, mostly in the form of posttranslational modifications. As information on posttranslational modifications is generally not available with k_{cat} *in vitro* data, we have to rely on the experimentalists choosing the most active form when conducting the biochemical assay. $k_{app,max}$ is based on a maximum across conditions, and individual $k_{app,max}$ values are thus not condition-specific. Nevertheless, it might be possible to train growth condition-specific models for k_{app} , and the studies cited by the reviewer suggest that this is a promising avenue for future research. That being said, we see the aim of our study as providing a 'default' parameterization that is independent of growth conditions.

Actions: We now cite the previously missing articles on the application of machine learning in bacterial physiology and applications in synthetic biology and metabolic engineering (l. 65-68). We further added the possibility of posttranslational modifications affecting *in vitro* assays (l. 302-304, also see l. 48-51) and the possibility of training condition-specific models (l. 361-362) to the discussion.

2. In Figure 2, cross-validated machine learning was based on R square. The R square for k-cat was up to 0.75 (relatively poor prediction quality). How many runs were used for each method to calculate cross-validated R²? If RMSE was used to compare performance, would authors have same conclusions about machine learning models?

Response: We used repeated 5-fold cross validation with 5 repetitions, leading to 25 error estimates for each model, except for the case of deep learning where we used a single iteration of 5-fold cross-validation (see Methods).

We agree that RMSE is an important measure of model performance that is not necessarily consistent with R². We thus added a summary of the cross-validated RMSE across models as

Supplementary Figure 3. They show the same performance hierarchy as our analysis with R^2 . Also note that we now, in addition to the cross-validation error, include a test set error. Finally, we'd like to note that we consider an R^2 of around 0.75 on a test set a surprisingly good model performance given the expected noise levels that underlie the biological data that the model is trained and tested on.

Actions: Added Supplementary Figure 3 that shows the RMSE for cross validation and test set across models (referenced in l. 125-127).

3. It would be of great value to see the performance of the models with other E. coli genome-scale metabolic network reconstructions. Particularly, the latest model, iML1515 (Nature Biotechnology volume 35, pages 904–908 (2017)).

Response: We agree that the newest version of the genome-scale reconstruction should be used (iML1515 was not published when we conducted our analyses).

Actions: We now updated all analysis to make use of iML1515. This includes all database extractions for structural features and kinetic parameters, re-calculation of flux sampling for the “flux” feature, all machine learning model training and analysis, and MOMENT simulations. All figures, except for Figure 1, were updated accordingly. This update does not affect the conclusions of our study.

4. The authors do not present a learning curve (graph that compares the training and cross validation error over a varying number of training instances) to see how the performance of these models depend on the size of the datasets (which are quite small). It is critical to understand how the accuracies would vary with the amount of data available.

Response: We agree that learning curves are a valuable tool for the evaluating model bias and the potential benefits of attaining additional data. We now computed learning curves for the elastic net and random forest models and added them as Supplementary Figure 4. As expected, the elastic net shows a higher bias than the more complex random forest model. Furthermore, the learning curves of the random forest for $k_{app,max}$ show that, while our data set size is in the region of diminishing returns, additional data has the promise of slightly improving performance.

Actions: Added Supplementary Figure 4 with learning curves for the elastic net and random forest models. Referenced the Figure in lines 187-188 and 354-358.

5. A good number of the features involved a lot of computations and could potentially be sources of error. The robustness of the machine learning models to these errors should be quantified.

Response: The most important feature in both our models of k_{cat} *in vitro* and $k_{app,max}$ is the “flux” feature. At the same time, as the reviewer points out, it is associated with a high uncertainty, as it was derived from *in silico* flux across random conditions. We now added a sensitivity analysis, where experimental flux data (from MFA) is used instead of MOMENT sampled flux. We show

that the models' performance (as measured by RMSE and R^2 on validation and test sets) does not change. Furthermore, in response to a concern by Reviewer 3, we now use parsimonious FBA as the 'default' flux feature, which likewise does not change the models' performance. We also now replaced our former importance metrics (presented in Figure 3) with a permutation importance, where the effect of randomly permuting a feature on model performance is quantified.

Finally, we note that our update of the modelling pipeline to use $\mathcal{M}L1515$ (described above) involved a recalculation of all features with the novel gene-protein-reaction rules of $\mathcal{M}L1515$. The fact that this change did not affect model performances supports the notion of high robustness of our models.

Actions: Added Supplementary Figure 5, showing model performance when MFA data is used to generate the flux feature. Feature permutation importances and associated p -values are presented in Figure 3.

Reviewer #3 (Remarks to the Author):

This work describes a machine learning based workflow to estimate enzyme catalytic activity. The authors have highlighted the key mechanistic features to train different machine learning models and have also applied the results from this study to parameterize two different protein cost models (i.e. MOMENT and ME model). Their results show improved quantitative proteome predictions by using the parameterized apparent turnover number (i.e. $k_{app,max}$). However, the authors did not clearly state the source of their training dataset, and their model features, model testing metric, and the comparisons with existing procedures are not thoroughly convincing as explained below.

Major Concerns

- MOMENT requires $k_{app,max}$ values to estimate fluxes. The authors mention that the $k_{app,max}$ values used in the training dataset for the Machine learning (ML) models were calculated using MOMENT. It is not clear which fluxes/datasets were used to estimate the training dataset. The authors have also mentioned that they used MOMENT to test their ML model predictions and observed improved predictions. Thus, it appears that the testing was better than training while using data from the same source. Could the authors comment on this?

Response: Firstly, to avoid any misunderstandings: MOMENT was not used to calculate the $k_{app,max}$ values in the training set. Davidi et al. (2016), the source of our $k_{app,max}$ data, used parsimonious FBA (pFBA), a method that does not require information on enzyme kinetics, to calculate $k_{app,max}$.

MOMENT requires estimates of k_{cat} or k_{app} for its predictions, and we utilized the default parameterization in the sybilccFBA package that is based on *in vitro* data for the calculation of the "flux" feature in the training data. We agree that this use of *in vitro* data is potentially problematic as some of the output data would be used to compute features, leading to overly optimistic performance estimates ("data leakage"). To solve this problem, we now use pFBA to compute the "flux" feature, showing that doing so does not decrease model performance (see

updated Figures 2 and 4) and that data leakage was not a problem in previous analyses. In addition to this use of pFBA, we also added a new analysis that makes use of experimental metabolic flux analysis (MFA) data across different growth conditions for computing the “flux” feature (see also our response to the Reviewer’s comment below). This MFA-based flux feature yields very similar performance compared to the two *in silico* flux features described above (Supplementary Figure 5), thus further supporting our finding that the models are robust to the source of the flux feature.

Actions: Replaced the MOMENT-based flux feature with a new pFBA-based flux feature in all analyses, updating all figures in the main text and the supplementary material. Added Supplementary Figure 5 that shows the effect of using MFA-based flux as a model feature on model performance.

- The authors have used the average of flux distributions, under varying nutrient conditions estimated using MOMENT, as a feature during model training. However, this feature does not represent a meaningful flux distribution in *E. coli*. The reviewer would instead suggest that fluxes estimated using ¹³C-MFA data can be used as a feature in the training data.

Response: We agree that the MFA data is a more reliable source for flux estimates. In our initial analysis we nevertheless decided to use *in silico* flux for three reasons: (1) MFA data has a low coverage that is normally far from genome-scale. (2) enzyme utilization in natural conditions is likely shaped by a high number of complex growth conditions. This complex selection pressure can be approximated by sampling *in silico* flux across randomly chosen environments, whereas MFA data will only be available for a small number of non-complex growth conditions. (3) MOMENT predictions were shown to exhibit a decent agreement with experimental flux data ($R=0.76$, (Adadi et al. 2012)).

That being said, we understand the Reviewer’s concern and thus used MFA data across 8 conditions to compute the flux feature. To alleviate the coverage problem of MFA data mentioned above, we extrapolate the MFA data by constraining an FBA problem to identify genome-scale steady-state flux distributions (see updated Methods). We show that using this MFA-based flux feature in our machine learning framework yields very similar performance when compared to the *in silico* features (pFBA and MOMENT), and we now present this result in Supplementary Figure 5.

Actions: Added Supplementary Figure 5 and described the procedure in the Methods section. The result is referenced in l. 153-155 and l. 308-310, and a new section explaining the procedure was added to the Methods section.

- The authors have reported that their *kcat in vitro* predictions had an average R^2 value of 0.22 compared to a value of 0.75 for $k_{app,max}$ predictions. The reviewer is wondering if the authors could improve the *kcat in vitro* further by using more features for training. It is also not convincing that the features are “important” due to low prediction fidelity of the *kcat in vitro* model. It is highly probable that that the “important” features would vary when the accuracy of machine learning model of *kcat in vitro* is improved.

Response: We agree that the high noise in k_{cat} *in vitro* data is a problem for inferring relevant features. We now try to counteract this problem by (1) taking measures to improve prediction performance and (2) by a more rigorous measurement of feature importance:

(1) In order to improve the performance of k_{cat} *in vitro* models, we now added a further step of manual curation to our data extraction pipeline for k_{cat} *in vitro*, where we manually select the most appropriate literature source(s) (focusing on *in vivo*-like conditions, recency of the study, and agreement among values) if multiple sources are available, making use of the Uniprot resource. Furthermore, we include additional features that are potentially predictive for k_{cat} *in vitro*, namely Michaelis constants, substrate and product concentrations, and thermodynamic efficiency of the reaction (as requested by the Reviewer below). These efforts increased the average cross-validated R^2 across models from 0.22 to 0.31 (now shown in Figure 2). Thus, while we were able to improve the performance of the k_{cat} *in vitro* models, it still lacks behind the models for $k_{app,max}$.

(2) The significant similarity in feature importance between the model for k_{cat} *in vitro* and the model for $k_{app,max}$ indicates that, even though the performance of the *in vitro* model is lower than the *in vivo* model, meaningful features can be identified. To support this idea, we now use a permutation-based feature importance that avoids a possible bias in feature complexity (Strobl et al. 2007) to analyze model-similarity and find that our previous results hold, except for the case of “EC number = 3” which was now assigned a higher importance in both models. Furthermore, we use a permutation test to calculate p -values for importances that allow us to focus on statistically significant contributions to model predictions.

In summary, we were able to yield a slight improvement in *in vitro* model performance and strengthened our analysis of model-similarity. While these results give us confidence that the most influential features for k_{cat} *in vitro* were correctly identified by the machine learning model, we agree with the reviewer that feature importance should be interpreted carefully in light of the model performance. We thus updated our discussion of *in vitro* features to only focus on the most important examples. Furthermore, (related to the Reviewer’s comment on our partial dependence analysis below) we now refrain from comparing model responses for the $k_{app,max}$ model to those in the k_{cat} *in vitro* model, except for the very clearly important and significant features, flux and generalist. We also note in the discussion that the low performance of the k_{cat} *in vitro* model warrants careful interpretation of the model’s feature importances.

Actions: We updated k_{cat} *in vitro* data set extraction and model training (described in l. 530-532). We re-worked Figure 3 by using permutation importance and permutation tests. Furthermore, we re-worked discussion of similarities between $k_{app,max}$ and k_{cat} *in vitro* ML models (l. 261-272, l.296-297).

- A few important features are missing in the feature selection section of the ML models. $k_{app,max}$ varies as a function of metabolite concentrations, ΔG , and k_m based on the study by Noor et al. [1]. However, the authors did not use these parameters as features for their machine

learning models. There are few studies [2, 3] that predict k_m using machine learning approaches. Thus, the features that correlate with k_m can also be included to contribute towards $k_{app,max}$ predictions. In addition, a recently published metabolomics dataset [4] can also be included in the feature set for metabolite concentrations.

Response: We agree that the effect of the mentioned features is of interest. We now include data on substrate and product concentrations, thermodynamic efficiency (a function of ΔG), and data on K_m *in vitro* as additional features. Surprisingly, we find that these features do not contribute to the predictions of $k_{app,max}$, but that K_m is the second most important feature for k_{cat} *in vitro*. As k_{cat} *in vitro* is predicted to increase with K_m , while the expected bias in $k_{app,max}$ has the opposite effect (consistent under-saturation will result in a $k_{app,max}$ that underestimates k_{cat}), it is possible that these contrasting effects lead to the low importance of K_m in models for $k_{app,max}$. We further thank the Reviewer for pointing out the studies on enzyme affinity prediction, which we now cite in the introduction as examples for successful use of machine learning techniques for the prediction of enzyme parameters.

Actions: Added the requested features and updated all figures accordingly and explained their use in lines 95-96 and in the Methods section. The effect of K_m is discussed in lines 159-161 and lines 284-287. Added citations to the introduction in lines 65-67.

- The accuracy of all the machine learning models are quantified using cross-validation R^2 values in Figure 2. These methods exhibit good accuracies (>75% in cross validation), but it is still possible that they may not perform well in the predictions (see SVM model in study [3]). An unbiased set of data should be used to clearly quantify the accuracy of prediction in this workflow (i.e. partition the dataset into train, validate, and test datasets).

Response: Based on the small size of the data set, we originally chose to estimate model performance solely based on its cross-validated error (i.e., on the validation set), but we agree that this approach has a possible optimistic bias due to hyperparameter optimization. Thus, we now kept aside testing sets and report the error on the unseen test set alongside the distribution of cross-validation error based on the remaining training set samples. We find that test set performances are close to the average cross-validation estimates, and we conclude that optimistic bias of cross-validation error was not a problem for our performance estimates. It should be noted that the variance in test set errors is expected to be high, as the underlying data sets are small. This effect leads to cases where the test error is lower than the median cross-validation error.

Actions: Updated Figure 2 and the new Figure 3 and 5 with test set errors, described in lines 127-129 and in the caption of Figure 2.

- The authors did not explain why the $k_{app,max}$ -based model has a better prediction than the model using the k_{eff} dataset from Ebrahim et al. [5]. The set of k_{eff} from Ebrahim et al. was fitted by minimizing the difference between model predicted proteome and experimentally measured proteome using the data from Schmidt et al. [6]. Theoretically, this set of k_{eff} should

give the best prediction (because the objective is similar to minimization of RMSE in Figure 4). It is surprising that kapp,max-based model can improve the already minimized value obtained from the constraint optimization problem by Ebrahim et al. [5].

Response: Like the Reviewer, we were surprised to find that the set of k_{eff} s provided by Ebrahim et al. does not yield a better performance. This effect is likely due to the procedure that Ebrahim et al. used for fitting the data, and the goals of the study for which their analysis was conducted. Firstly, Ebrahim et al. did not minimize absolute error for protein abundance on log scale, thus prioritizing the prediction of highly expressed proteins (as abundances are approximately log-normal distributed). Furthermore, the iterative algorithm does not guarantee convergence and is prone to finding local optima that were then averaged. Finally, Ebrahim et al. focused on showing condition-independent behaviour in k_{eff} s. Thus, they used a consensus set of inferred 284 keffs that showed a low condition-dependence. This leaves 1000+ keffs that they predicted based on a SASA, akin to only using the molecular weight feature in our modelling framework. The feature importance estimates we present in Figure 3 suggest that using SASA alone is unlikely to yield accurate predictions, indicating why the performance of these k_{eff} vectors is low.

Action: We added an explanation to the discussion (l. 331-335).

Minor Concerns

- As mentioned by the author in line 270-271, protein cost model by using parameters trained in this machine learning model can only provide an upper limit because effective enzyme turnover rate is condition-dependent. Is it possible to include more machine learning features (based on each condition) to obtain a more accurate model for each condition?

Response: While the focus of our study was to provide a 'default' parameterization that is independent of growth conditions, it is certainly promising to study the condition-specific structure in experimental k_{app} vectors. Machine learning studies that aimed at predicting *E. coli* condition-specific physiology (Kim et al. 2016; Wu et al. 2016) suggest that this could be a promising approach.

Action: We now mention the promise of context-specific models in the discussion (l. 361-362).

- Line 197 to 215: the last section seems redundant to the reader because the authors have already shown that similar feature importance exists between ML models of kapp,max and kcat. It is expected that an ensemble of ML models will have consistent important features. Can the authors comment on this?

Response: Our original motivation for the inclusion of the partial dependence analysis in a prominent position of the results section was that it allows us to compare the shape of variable responses learned by the two models, which is not possible based on the importances alone. We agree with the reviewer that this might seem redundant to the reader, and, especially in the light of the valid criticism about the questionable importance of features in the k_{cat} *in vitro* model (see point 3 above), we have decided to rework our partial dependence analysis and to transfer

it to the supplementary data. We now only show the most important features for each model that were also found to be significant by permutation testing and shorten its discussion to avoid redundancy.

Actions: Reworked partial dependence analysis and transferred it to the supplementary material. Feature responses are now compared in the discussion (l. 267-277).

- In page 9 line 212, the authors should replace "...that only show only small..." with "...that show only small ...".

Response: Sentence was removed.

References:

1. Noor, E., et al., A note on the kinetics of enzyme action: A decomposition that highlights thermodynamic effects. *Febs Letters*, 2013. 587(17): p. 2772-2777.
2. Carbonell, P. and J.L. Faulon, Molecular signatures-based prediction of enzyme promiscuity. *Bioinformatics*, 2010. 26(16): p. 2012-9.
3. Mellor, J., et al., Semisupervised Gaussian Process for Automated Enzyme Search. *ACS Synth Biol*, 2016. 5(6): p. 518-28.
4. McCloskey, D., et al., RapidRIP quantifies the intracellular metabolome of 7 industrial strains of *E. coli*. *Metab Eng*, 2018. 47: p. 383-392.
5. Ebrahim, A., et al., Multi-omic data integration enables discovery of hidden biological regularities. *Nat Commun*, 2016. 7: p. 13091.
6. Schmidt, A., et al., The quantitative and condition-dependent *Escherichia coli* proteome. *Nat Biotechnol*, 2016. 34(1): p. 104-10.

Additional changes

- "Supplemental" prefix is now "Supplementary" to meet formatting guidelines (this change is not marked for the sake of facility of inspection).
- Added citation of Davidi et al. (2018) (l. 282-284).
- Added explanatory sentence to the description of Figure 4.
- Replaced bar plots that represented distributions with box plots as required by journal presentation guidelines (relevant for Figure 2, Supplementary Figure 3, and Supplementary Figure 5).
- Added information on the treatment of membrane proteins to the Methods section.
- Made use of tenses in the Results section consistent and added minor stylistic improvements.

References

- Adadi, Roi, Benjamin Volkmer, Ron Milo, Matthias Heinemann, and Tomer Shlomi. 2012. "Prediction of Microbial Growth Rate versus Biomass Yield by a Metabolic Network with Kinetic Parameters." *PLoS Computational Biology* 8 (7): e1002575–e1002575.
- Davidi, Dan, Liam M. Longo, Jagoda Jabłońska, Ron Milo, and Dan S. Tawfik. 2018. "A Bird's-Eye View of Enzyme Evolution: Chemical, Physicochemical, and Physiological Considerations." *Chemical Reviews*, August. <https://doi.org/10.1021/acs.chemrev.8b00039>.
- Davidi, Dan, Elad Noor, Wolfram Liebermeister, Arren Bar-Even, Avi Flamholz, Katja Tummler, Uri Barenholz, Miki Goldenfeld, Tomer Shlomi, and Ron Milo. 2016. "Global Characterization of in Vivo Enzyme Catalytic Rates and Their Correspondence to in Vitro Kcat Measurements." *Proceedings of the National Academy of Sciences* 113 (12): 3401–6.
- Kim, Minseung, Navneet Rai, Violeta Zorraquino, and Ilias Tagkopoulos. 2016. "Multi-Omics Integration Accurately Predicts Cellular State in Unexplored Conditions for Escherichia Coli." *Nature Communications* 7.
- Strobl, Carolin, Anne-Laure Boulesteix, Achim Zeileis, and Torsten Hothorn. 2007. "Bias in Random Forest Variable Importance Measures: Illustrations, Sources and a Solution." *BMC Bioinformatics* 8 (January): 25.
- Wu, Stephen Gang, Yuxuan Wang, Wu Jiang, Tolutola Oyetunde, Ruilian Yao, Xuehong Zhang, Kazuyuki Shimizu, Yinjie J. Tang, and Forrest Sheng Bao. 2016. "Rapid Prediction of Bacterial Heterotrophic Fluxomics Using Machine Learning and Constraint Programming." *PLoS Computational Biology* 12 (4): e1004838.

REVIEWERS' COMMENTS:

Reviewer #2 (Remarks to the Author):

Authors have addressed all my comments.

Reviewer #3 (Remarks to the Author):

Satisfied with the responses and changes to the manuscript. No further comments.